# OpenReview forum: "SpreadsheetBench: Towards Challenging Real World Spreadsheet Manipulation"
_NeurIPS.cc/2024/Datasets_and_Benchmarks_Track — NeurIPS 2024 Track Datasets and Benchmarks Spotlight_

### Official Review · Reviewer_VgK3 · 2024-06-26
**Review of submission 2435**

**Rating:** 7
**Confidence:** 3

**Review:**

Overall, the paper clearly explains the procedure involved in assembling, cleaning and preparing the questions and examples in the dataset. Human annotators are heavily involved in the procedure. The experimental section makes sense, and clearly highlights that the benchmark dataset contains hard questions and that the currently available methods do not produce acceptable results.

Opportunities for improvement include 1) clarifying whether the dataset is truly software-agnostic, or if it is Excel-only; 2) expanding the experimental section with more detail on how the LLMs interface with the benchmark, and how the different LLMs treat spreadsheet-specific features (formatting, multi-table sheets etc.); 3) describing the role of annotators in more detail.

**Strengths:**

- The benchmarking dataset introduced with the submission markedly improves prior work.
- The dataset is constructed in a sound way and includes hard problems that can stress the capabilities of LLMs in different ways. The preparation procedure is also explained clearly.
- The experimental section provides good insight in some of the limitations of various LLMs and highlights how spreadsheet manipulation remains a difficult problem for current LLMs.

**Additional Feedback:**

There are some typos.

In figure 1, it is not clear from the example if the description (e.g., "Non-standard table (Nested Header)") is a part of the spreadsheet passed to the LLM, or if it's a label used exclusively for the sake of the example. In the caption, it might be easier to replace the color names simply by coloring the text appropriately ("output example" would be red).

**Clarity:**

The paper is clear and understandable, however it could be improved by cleaning up the examples. It would also help to clarify with an example how the LLMs are acting on the spreadsheets, or whether any interaction is done internally by the LLM and only the result is given (or if this is LLM-dependent).
Providing examples of the interface and/or procedure required to test a LLM with a given question would help.

**Correctness:**

The benchmark processing steps make sense, and the overall work is convincing. However, it is not clear whether all tests have been evaluated on multiple spreadsheet manipulation software, or if experiments have been run exclusively with Excel.

**Documentation:**

The documentation presented in the readme and the website looks exhaustive, although part of the code may only be run on Windows. A sample of the benchmarking dataset and the corresponding results are available for download. No licensing information is provided, and the raw data (a list of the used forum/blog posts) is not available.

**Ethics:**

I believe that ethical concerns have been addressed well in the appendix.

**Limitations:**

Some of the limitations in data selection and test case construction are evaluated in the additional material. Additionally:
- Qualitative analysis of the LLMs is lacking; it would be interesting to have more insight into how different LLMs interact with different characteristics of a spreadsheet (e.g., whether a specific model struggles more with formatting, or multiple tables etc.).
- Although the benchmark is advertised as "software-agnostic", from the text it is not clear whether it was tested to guarantee it (i.e., the same tests have been run on software other than Excel). This point should be clarified.

**Opportunities For Improvement:**

- Spreadsheet manipulation is not inherently software-independent, as some functions may not be available (or may change name) changing software. This is also reflected in the linked repository, as it is supposed to run exclusively on Windows, and all available spreadsheets are in xlsx format. It is unclear from the text whether the examples have been tested on a variety of spreadsheet manipulation software, or exclusively on Excel. If no verification of additional software was performed, then the benchmark is still "tested only on Excel", and should be clarified that this is the case. Potential future work could involve this verification step and, possibly, evaluating how software-specific functions or features affect LLMs.
- Annotators contributed significantly to the submission, however almost no information is provided about them in the main body of the paper. Details about the number and preparation of the annotators should be available in the main body.
- There should be more detail about the interface that is used feed test spreadsheets to each LLM, and more insight on the capabilities of different LLM models. For example, can all LLMs parse formatting (color, bold etc.) in the same way? Are some of them "blind" to formatting? Are there other model-specific limitations that differentiate the LLMs, other than the performance measured using the metric? It would be useful to have an example of the output of the LLM for a given test set.

**Relation To Prior Work:**

Prior work is well explored and various baselines for LLMs are evaluated.

**Summary And Contributions:**

The authors introduce SpreadsheetBench, a dataset of questions and examples designed to evaluate how effective LLMs are at performing various spreadsheet manipulation operations. Questions gathered from online Excel forums, which have been cleaned and prepared by human annotators. Results suggest that the questions reported in the benchmark are hard for LLMs to solve, and that further work on spreadsheet manipulation is needed.

---

> ### Author Rebuttal · Authors · 2024-08-16
>
> We appreciate your detailed and insightful reviews on our manuscript, additional resources and Github repo! We are encouraged that you find our submission markedly improves prior work, constructed in a sound way, and provides good insight in the experimental section. We would like to address your questions in the following part and include these details in our next revision.
>
> > **Three similar comments about software-independent:**
> > 1. Comment A in Opportunities For Improvement: Spreadsheet manipulation is not inherently software-independent, as some functions may not be available (or may change name) changing software. This is also reflected in the linked repository, as it is supposed to run exclusively on Windows, and all available spreadsheets are in xlsx format. It is unclear from the text whether the examples have been tested on a variety of spreadsheet manipulation software, or exclusively on Excel. If no verification of additional software was performed, then the benchmark is still "tested only on Excel", and should be clarified that this is the case. Potential future work could involve this verification step and, possibly, evaluating how software-specific functions or features affect LLMs.
> > 2. Comment B in Limitations: Although the benchmark is advertised as "software-agnostic", from the text it is not clear whether it was tested to guarantee it (i.e., the same tests have been run on software other than Excel). This point should be clarified.
> > 3. Comment in Correctness: The benchmark processing steps make sense, and the overall work is convincing. However, it is not clear whether all tests have been evaluated on multiple spreadsheet manipulation software, or if experiments have been run exclusively with Excel.
>
> That's a great question! We employ the following methods to ensure software independence as much as possible:
>
> 1. **Software-independent instruction:** Not all spreadsheet manipulation instructions extracted from Excel forums are software-independent, as some instructions may involve special components (e.g., “input boxes,"  "buttons,"  and “forms”) or advanced charts that only appear in Excel. To minimize software dependence on the instructions, we only select questions that focus on the manipulation of the value of cells in the spreadsheet and try to avoid questions involving special components or advanced operations, as demonstrated in line 122 of the main body and the Figure 12, Figure 13 in Appendix.
> 2. **Software-independent spreadsheet files:** It's true that all the spreadsheet files in our dataset are in the XLSX format. This file format is widely supported across multiple spreadsheet applications, such as Microsoft Excel, Google Sheets, LibreOffice Calc, and WPS Office. Therefore, the XLSX format itself is not considered software-dependent. However, it's worth noting that some spreadsheet software may not provide full support for the XLSX format. For example, certain advanced components and charts may not display properly in applications other than Excel. During our instruction filtering process, we have made an effort to exclude any content that may be incompatible with non-Excel software.
> 3. **Software-independent LLM inference:** In our approach, we require the Language Models (LLMs) to produce code solutions that manipulate spreadsheets, as demonstrated in line 97 of the main body. Instead of utilizing TypeScript and the Excel API to interact with spreadsheet files[1], the LLMs generate Python code that manipulates the spreadsheets using software-independent libraries such as pandas and openpyxl. Furthermore, when evaluating the correctness of the code solutions, we compare the values of the cells in the spreadsheet, rather than assessing the similarity of the generated TypeScript code[1]. Specifically, we automatically opens the spreadsheet files after manipulation and the spreadsheet files with ground truth **using pandas and openpyxl libraries**, and compare the values of the cells to see if they are consistent, which is still software and operating system independent. This ensures that our inference process is not dependent on any specific spreadsheet software or API.
>
> A specific situation arises when the code solution generated by the LLM includes the insertion of a formula into the spreadsheet files. In such cases, we need to first open the spreadsheet using a spreadsheet software  to calculate the result of the formula before obtaining the ground truth cell values. Manually opening each spreadsheet file individually would be inefficient. Therefore, we use the win2com library in Windows to automatically open each spreadsheet file before conducting the exact match. The win2com library currently only supports Excel and WPS Office softwares, resulting in a dependence of the current evaluation process on Windows, Excel, or WPS Office.
>
> Overall, our benchmark for LLM inference on intructions and spreadsheets is software-independent. However, for a fully automated evaluation, we now run the evaluation process under Windows using Excel or WPS Office software. We will make efforts in the future to find a thoroughly automated software-independent evaluation method that addresses formula-related challenges.
>
> *Reference:*
>
> [1] Payan J, Mishra S, Singh M, et al. InstructExcel: A Benchmark for Natural Language Instruction in Excel[C]//Findings of the Association for Computational Linguistics: EMNLP 2023. 2023: 4026-4043.

---

> > ### Author Rebuttal · Authors · 2024-08-16
> >
> > > **Two similar comments about examples or interface of the evaluation:**
> > > 1. Comment C in Opportunities For Improvement: There should be more detail about the interface that is used feed test spreadsheets to each LLM.
> > > 2. Comment in Clarity: The paper is clear and understandable, however it could be improved by cleaning up the examples. It would also help to clarify with an example how the LLMs are acting on the spreadsheets, or whether any interaction is done internally by the LLM and only the result is given (or if this is LLM-dependent). Providing examples of the interface and/or procedure required to test a LLM with a given question would help.
> >
> > Thanks for your valuable comments. We will give a clearer explanation of our evaluation process and provide an example in the attached PDF.
> >
> > We utilize two distinct settings (i.e., single round and multi-round) as described in line 245 in the main body.
> >
> > - Single Round: We present the model with the initial few rows of spreadsheet files within the prompt, allowing for only one inference. The prompt is provided in Figure 19 in Appendix.
> > - Multi-Round: Building on the single-round prompt setting, we incorporate an additional prompt that allows LLM to actively obtain the information in the spreadsheet. Specifically, we offer LLMs two choices: to generate code for retrieving spreadsheet contents (e.g., text value, color, font style) or to provide the ultimate solution. If the model asks for the content of spreadsheet, we will provide the execution result in the next round of the conversation. The prompt is provided in Figure 20 in Appendix.
> >
> > We provide an example of the response of GPT-4o in multi-round setting, as shown in the attached PDF file. We first conduct a multi-turn conversation to obtain a code solution, before applying the solution to the target spreadsheet.
> > - **Code Solution Generation:** First, we send the instruction and first few rows of the spreadsheet to LLM. Then, LLM generate code to obtain the value of the column named "Data". Third, we execute the code generated by LLM and provide the table values to the LLM. Fourth, based on the dialogue history and the table values provided by the code executor, LLM gives the final code solution. Finally, after the error checking by the executor, this code solution is used as the final result of this test case.
> > - **Code Solution Execution:** The code solution is automatically applied to the target spreadsheet by our Python script. Comparing the execution result with the value in the ground truth spreadsheet, the code solution is passed on this test case.

---

> > > ### Author Rebuttal · Authors · 2024-08-16
> > >
> > > > **Two similar comments about qualitative analysis:**
> > > > 1. Comment C in Opportunities For Improvement: There should be more insight on the capabilities of different LLM models. For example, can all LLMs parse formatting (color, bold etc.) in the same way? Are some of them "blind" to formatting? Are there other model-specific limitations that differentiate the LLMs, other than the performance measured using the metric? It would be useful to have an example of the output of the LLM for a given test set.
> > > > 2. Comment A in Limitations: Qualitative analysis of the LLMs is lacking; it would be interesting to have more insight into how different LLMs interact with different characteristics of a spreadsheet (e.g., whether a specific model struggles more with formatting, or multiple tables etc.).
> > >
> > > Thanks for your insightful comments! We provide two cases in the attached PDF and conduct qualitative analysis of the LLM's performance on different characteristics of the spreadsheet. To get the code solution of these two cases, we utilize the multi-round setting that is discussed in the previous questions.
> > >
> > > **Example 1** shows an instruction that manipulates a non-standard relational table with formatting in spreadsheet. Both GPT-3.5 and GPT-4 understood the question and read the spreadsheet properly. However, due to the lack of coding ability, GPT-3.5 uses the wrong element "correct_results[cell.column]" (marked as red). The correct one should be "correct_results[cell.column - 1].value". GPT-4o can generate the correct Python code, highlighting the importance of coding ability for the accuracy of spreadsheet manipulation.
> > >
> > > **Example 2** shows an instruction that manipulates two tables in one sheet with formatting. The two GPT-4 inference results both show issues with understanding the structure of the spreadsheet.
> > >
> > > - For the first result of GPT-4o, misalignment occured when reading the tables. "5A1" belong to the column names "Green" but is highlighted in yellow. "5A3" belong to the column names "Amber" but is highlighted in red. The wrong code snippets are highlighted in red.
> > > - For the second result of GPT-4o, the model did not read the whole lookup table in column E to G. The rows after the tenth row have been ignored. Thus, "1A4", "D2", and "1A1" are not highlighted. The wrong code snippets are highlighted in red.
> > >
> > > These two examples show that proficient coding skills and a solid understanding of various spreadsheet structures are essential for the spreadsheet manipulation task.

---

> > > > ### Author Rebuttal · Authors · 2024-08-16
> > > >
> > > > > Comment B in Opportunities For Improvement: Annotators contributed significantly to the submission, however almost no information is provided about them in the main body of the paper. Details about the number and preparation of the annotators should be available in the main body.
> > > >
> > > > Thanks for your comment. Due to the page limitation of the main body, we placed the introduction of annotators in Appendix B.2. In our next revision, we will adjust the content of the main body and add the following part about the annotation team in the main body:
> > > >
> > > > **Annotation Team.** We employ a team of 20 individuals who specialize in Excel and have extensive experience in annotation. All team members hold a bachelor’s degree and are compensated in line with market rates. For data annotation, we allocate a predetermined quantity of data to annotators on a daily basis, and the allocation is modified in accordance with local regulations regarding work hours. For data validation, we hire two experienced validators with bachelor’s degrees for the first round of quality checks. Moreover, two authors who hold master’s degrees perform a secondary round of quality assessments to guarantee the high quality of data.
> > > >
> > > > > Comment in Documentation: No licensing information is provided, and the raw data (a list of the used forum/blog posts) is not available.
> > > >
> > > > Thanks for your comments. We have provided the license information in our Github repo, which is [CC BY SA 4.0](https://creativecommons.org/licenses/by-sa/4.0/). We will add this license information in the Appendix of our manuscript in the next revision.
> > > >
> > > > For the raw data, as described in Appendix A.3 (3), we will refrain from engaging in any secondary distribution of the raw data, considering potential licensing and copyright risks.
> > > >
> > > > > Comments in Additional Feedback: There are some typos. In figure 1, it is not clear from the example if the description (e.g., "Non-standard table (Nested Header)") is a part of the spreadsheet passed to the LLM, or if it's a label used exclusively for the sake of the example. In the caption, it might be easier to replace the color names simply by coloring the text appropriately ("output example" would be red).
> > > >
> > > > Thanks for your valuable comments on the typos of our manuscript! The four types of tables in Figure 1 are manually summarized based on the actual tables in the spreadsheet files, and these types (e.g., "Non-standard table (Nested Header)") will not be sent to the LLMs during the inference process. In our next revision, we will provide a clear explanation in the Figure 1 caption.
> > > >
> > > > Replacing the color names simply by coloring the text appropriately is a good idea and make the caption easy to read! We will make this modification in our next revision.

---

> > > > > ### Comment · Reviewer_VgK3 · 2024-08-19
> > > > >
> > > > > Thanks for the exhaustive replies and for addressing my concerns. I have no further comments.

---

> ### Comment · Reviewer_VgK3 · 2024-08-19
>
> Thanks for the detailed response, it addresses well my concerns about software independence. I believe that adding to the main body (even in a shortened form) and to the appendix the information about how software independence is ensured would further convince readers of the validity of the work, and ease reproducibility.
>
> In general, how the LLMs interacted with the spreadsheets was not clear at all from the text, I believe it is important to clarify that the LLMs are in fact acting on the spreadsheets using a code interface. This would be helpful especially for readers that are not familiar with the direct capabilities of the models involved.

---

> > ### Author Response · Authors · 2024-08-19
> >
> > Thanks again for your valuable review and comments. In our next revision, we will incorporate the explanation about software-independence into both the main text and the appendix of our manuscript. We will also give a clearer explanation of the code interface between LLMs and our benchmark by adding textual information to the main body and adding figure examples to the appendix.

---

### Official Review · Reviewer_Co7Y · 2024-07-03
**SpreadSheetBench,  a benchmark to evaluate the immersion of current LLMs in the workflow of spreadsheet users.**

**Rating:** 8
**Confidence:** 5
**Clarity:** Paper is well written.

**Review:**

This paper presents a benchmark (SpreadsheetBench) to evaluate the immersion of current LLMs in the workflow of spreadsheet users. The SpreadsheetBench is built from 912 real questions gathered from online Excel forums reflecting the need of real users. This is different from other benchmarks in the sense that they used synthetic queries for evaluation purposes. Also, the length of the questions/queries was substantially long in the SpreadsheetBench as opposed to other benchmarks. The paper discussed a number of evaluation methods revealing the substantial gap between the models and human performance.
The paper had a clear goal and the results. The results highlight the limitations (largely) of using LLMs in the workflow of spreadsheet users. This benchmark is beneficial to work towards improving the LLMs for spreadsheet use cases.

**Strengths:**

The submission is of high quality and very rigorous in several ways:

 - Benchmarking of the tools which exploits LLMs in day-to-day use cases is important to assess the accuracy of work done so far and understand the existing challenges

- The concept of benchmarking is still evolving and new methods are still investigated. So, new ways of benchmarking and considering the different possibilities in evaluating a tool is itself helpful towards establishing the concept of benchmarking. Therefore it is also beneficial for the broader research community in helping them think through the ways of benchmarking to evaluate the performance of LLMs and tools utilizing these LLMs.

**Additional Feedback:**

Authors:
1. Could you please point me to the discussion related to how the authors have ensured the data privacy and consent.
2. I do not see the maintenance plan for the code. Do you have any or if not please add this to the paper and to the GitHub repo.

**Correctness:**

The evaluation methods and experiment design looks appropriate and performed correctly. The results are presented in a way that it is easy to follow along.

**Documentation:**

The benchmark code, data and details are provided along with hosting, licensing plans. I could not find a maintenance plan.

**Ethics:**

The dataset used in this benchmark uses real questions gathered from real online forums.  This could be an ethical concern revealing the identification of users who published the questions. I do not see in the papers discussion related to how the authors have ensured the data privacy and consent.

**Limitations:**

Authors did not address the limitations and any societal impact of their work.

**Opportunities For Improvement:**

1. OJ style evaluation - please explain this in one line. Directly mentioning from the literature review is causing confusion and makes it hard to read the paper.
2. Please address the limitations and any societal impact of this work.

**Relation To Prior Work:**

The paper clearly explains the difference from previous contributions.

**Summary And Contributions:**

In this paper, authors have developed SpreadsheetBench comprising 912 instructions and 2,729 test cases, with an average of three test cases per instruction.  The instructions cover 10 primary manipulation categories and involve spreadsheets containing multiple tables extending beyond 100 columns and 20, 000 rows. Moreover, 35.7% of the spreadsheets contain multiple tables, and 42.7% feature 72 non-standard relational tables. This benchmark is unique due to the use of real-world instructions, diverse 73 spreadsheet formats, and a comprehensive testing strategy.

The evaluation of different models such as TableQA, open source LLMs for general and coding tasks, the closed source models, GPT 3.5 and GPT 4o and spreadsheet specific LLMs is performed to assess the performance. The authors use Online Judge style evaluation metric.

The paper has the following outcomes: there remains the scope of improvement of coding abilities within LLMs for spreadsheet manipulation and indicates that multi-round prompting, allowing LLMs to read spreadsheet data and error feedback from code compilers can increase the probability of correct responses.  The current LLMs and spreadsheet agents are inadequate in managing complex spreadsheet manipulation tasks as required by real-world scenarios.

---

> ### Author Rebuttal · Authors · 2024-08-16
>
> We appreciate your detailed and valuable reviews! We are encouraged that you find our work is of high quality and very rigorous. Your concerns are addressed as follows:
>
> > Comment A in Opportunities For Improvement: OJ style evaluation - please explain this in one line. Directly mentioning from the literature review is causing confusion and makes it hard to read the paper.
>
> Thank you for pointing this out. We apologize for the confusion caused by the description of OJ style evaluation.
>
> An Online Judge (OJ) style evaluation involves inputting multiple test cases into a code script and comparing each test case's output to the ground truth result. This method is more comprehensive and reliable than evaluating the code script just once. In our benchmark context, each test case is a spreadsheet file with a similar table structure but different cell values. This allows us to evaluate a code solution more thoroughly and determine its effectiveness in handling corner cases.
>
> > **Two similar comments about limitations and societal impact:**
> > 1. Comment B in Opportunities For Improvement: Please address the limitations and any societal impact of this work.
> > 2. Comment in Limitations: Authors did not address the limitations and any societal impact of their work.
>
> Thanks for your comment. Due to the page limitation of the main body, we have placed the limitation and societal impact sections in Appendices A.1 and A.2, respectively.
>
> - In limitation section (i.e., Appendix A.1), we present that our major limitations lie in data selection and the test case construction process. (1) For data selection, we eliminate posts that lack an acknowledged response or are difficult to formalize, which may also contain some valuable questions. (2) For test case construction, we did not meticulously devise every corner case for each question. However, we still request annotators to remain vigilant regarding potential corner cases while annotating each question.
> - In societal impact section (i.e., Appendix A.2), we demonstrate that our objective is to improve the proficiency of LLMs in comprehending and manipulating spreadsheets, with the ultimate aim of automating the spreadsheet manipulation process and reduce human workload in the future. Moreover, we also present that we exclude any content that could violate personal privacy, contain explicit material, depict violence, or involve other sensitive subjects during our human annotation process.
>
> Please refer to Appendices A.1 and A.2 for detailed information.
>
> > **Two similar comments about data privacy and consent:**
> > 1. Comment in Ethics: The dataset used in this benchmark uses real questions gathered from real online forums. This could be an ethical concern revealing the identification of users who published the questions. I do not see in the papers discussion related to how the authors have ensured the data privacy and consent.
> > 2. Comment A in Additional Feedback: Could you please point me to the discussion related to how the authors have ensured the data privacy and consent.
>
> Thanks for your question. We have discussed data privacy, data consent, and copyright in Appendices A.2 and A.3 (3).
>
> - In the process of raw data acquisition, to prevent copyright infringement, we follow the Robots Exclusion Protocol of the forum to crawl the content of the posters. We also prohibit any secondary distribution of raw data, considering potential licensing and copyright risks.
> - In the process of data annotation, we exclude any content that could violate personal privacy, contain explicit material, depict violence, or involve other sensitive subjects. Thus, we believe that the probability of our benchmark causing adverse effects on safety, security, discrimination, surveillance, deception, harassment, human rights, bias, and fairness is extremely minimal.
>
> Please refer to Appendices A.2 and A.3 (3) for detailed information.
>
> > **Two similar comments about maintenance plan:**
> > 1. Comment in Documentation: The benchmark code, data and details are provided along with hosting, licensing plans. I could not find a maintenance plan.
> > 2. Comment B in Additional Feedback: I do not see the maintenance plan for the code. Do you have any or if not please add this to the paper and to the GitHub repo.
>
> Thanks for your comment about missing maintenance plan! We have answered eight questions in the last section of the datasheet (i.e., Datasheet.pdf) about maintenance. However, we do agree that we need to organize this maintenance part and also place it in the appendix of the paper and the Github repo. Below is our maintenance plan and we will add this part in the next revision of our manuscript:
>
> RUC KBReasoning group will maintain this benchmark in the long term. We will continue to update the benchmark to fix labelling errors, instructions, or other necessary modifications, and all data will be versioned. We also welcome all contributors interested in our benchmark. If you have any questions or want to extend/augment/build on/contribute to the dataset, feel free to contact us via Github or E-mail (<zeyaoma@ruc.edu.cn>, <zbhmint@bit.edu.cn>).

---

> > ### Comment · Reviewer_Co7Y · 2024-08-16
> > **rebuttal reply**
> >
> > Thanks for the answers to my questions. I have no other questions however I have a comment (see below).
> >
> > I see that all of the information which I found missing in the paper is actually in the appendix. I suggest that you put the information in the paper as well. I understand that there is a limit on the number of pages but you can briefly put the limitation and societal impact of this work in the paper itself. This will help readers who read only the paper and draw conclusions from that.

---

> > > ### Author Rebuttal · Authors · 2024-08-16
> > >
> > > Thanks for your valuable comment and advice. In our next revision, we will summarize the limitations and societal impact of this work and put them in the main body of our paper.

---

### Official Review · Reviewer_wseN · 2024-07-05

**Rating:** 8
**Confidence:** 4
**Correctness:** Correct as far as I can tell.
**Clarity:** Clear.

**Review:**

The paper is well written and proposes an interesting and, to the best of my
knowledge, novel benchmark. In particular, the authors convincingly highlight
the shortcomings of existing benchmarks, which are much easier and not as
representative of real world scenarios. Overall, the paper is convincing.

There are some details of the benchmark and empirical evaluation that should be
clarified. How exactly was human performance measured here, in particular for
benchmarks that were not exactly the same as what was asked in forums? How
easy/difficult would it be to integrate new LLMs with the proposed benchmark?

**Strengths:**

- challenging and novel benchmark that highlights shortcomings of existing systems

**Additional Feedback:**

n/a

**Documentation:**

Sufficient as far as I can tell.

**Limitations:**

Yes.

**Opportunities For Improvement:**

See review.

**Relation To Prior Work:**

Yes.

**Summary And Contributions:**

The paper proposes a new benchmark for spreadsheet manipulation that includes
real world scenarios gathered from the web. The authors describe their benchmark
and run some experiments, demonstrating that the proposed benchmark is more
challenging than existing ones and highlights the shortcomings of existing
systems for LLM-based spreadsheet manipulation.

---

> ### Author Rebuttal · Authors · 2024-08-16
>
> Thank you for your valuable reviews! We are encouraged that you find our benchmark challenging and novel that highlights the shortcomings of existing systems. We would like to address your questions in the following part and make necessary modifications in our next revision.
>
> > Question A in Review: How exactly was human performance measured here, in particular for benchmarks that were not exactly the same as what was asked in forums?
>
> Thanks for your question. Due to the page limitation of the main body, we present the evaluation methodology of human performance in the second paragraph of Appendix C.2.
>
> To assess human performance, we did not directly use the responses from the original posts. Instead, we hired four expert Excel annotators to evaluate their performance on our benchmark using the provided instructions. The annotators were asked to supply formula solutions or VBA code to complete each instruction, rather than simply filling in the target answer. To minimize costs, we used a subset of fifty instructions and three test cases per instruction for human evaluation. This entire human assessment process was completed within a single day.
>
> We will add a clear reference to the evaluation methodology of human performance (i.e., Appendix C.2) in the main body, or move this paragraph to the main body if we have enough space.
>
> > Question B in Review: How easy/difficult would it be to integrate new LLMs with the proposed benchmark?
>
> Thank you for your question! It's easy to integrate new LLMs with the proposed benchmark.
>
> As described in Section 2.1, each data sample in our benchmark includes an instruction and a spreadsheet file as input. The output is a code solution for manipulating the spreadsheet. Users can choose either a single-turn or multi-turn LLM instruction approach to input the instruction and the spreadsheet file into the LLMs. The corresponding prompts are shown in Figures 19 and 20 in the Appendix. It is easy to integrate current new LLMs such as Llama3.1, deepseek-v2, etc. Users can design their own evaluation pipeline, as long as they input the instruction and spreadsheet content into the LLMs.

---

> > ### Comment · Reviewer_wseN · 2024-08-16
> >
> > Thank you for the clarifications.

---

### Official Review · Reviewer_35VV · 2024-07-21
**Interesting realistic benchmark**

**Rating:** 9
**Confidence:** 4
**Correctness:** Yes
**Clarity:** Yes

**Review:**

This is a well written paper, the dataset is thoughtfully constructed, it is realistic and is likely to become a valuable resource for the space of tabular data manipulation.

**Strengths:**

- well written
- well constructed
- realistic
- multiple test cases

**Additional Feedback:**

Have you addressed the licensing issues of repurposing these forums to create the dataset?

**Documentation:**

Yes

**Limitations:**

I may have missed it but I didnt see this section - however I do not know if there are any negative impacts.

**Opportunities For Improvement:**

Not sure I have many suggestions

**Relation To Prior Work:**

Yes

**Summary And Contributions:**

This paper describes the creation of a new spreadsheet benchmark which involves  912 instructions and 2,729
test cases, with an average of three test cases per instruction. Tables used in this benchmark are real, extracted from online spreadsheet forums, with sheets  containing multiple tables, and non-standard relational structures that are common in spreadsheet. In addition, the paper describes a careful construction of multiple test cases using human annotations from the forums for specific code.  The datasets are perturbed to try to mitigate the data leakage issues (since many neural models are trained on web data).

---

> ### Author Rebuttal · Authors · 2024-08-16
>
> We much appreciate your reviews and recognition of our work! We are encouraged that you find our benchmark realistic and valuable, and our manuscript well-written and well-constructed. Your concerns are addressed as follows:
>
> > Comment in Limitations: I may have missed it but I didnt see this section - however I do not know if there are any negative impacts.
>
> Thanks for your comment. Due to the page limitation of the main body, in our submitted version, we have placed the limitation section in Appendix A.1, where we present that our major limitations lie in data selection and the test case construction process.
>
> (1) For data selection, we eliminate posts that lack an acknowledged response or are difficult to formalize, which may also contain some valuable questions. (2) For test case construction, we did not meticulously devise every corner case for each question. However, we still request annotators to remain vigilant regarding potential corner cases while annotating each question.
>
> Please refer to Appendix A.1 for detailed content.
>
> > Question in Additional Feedback: Have you addressed the licensing issues of repurposing these forums to create the dataset?
>
> Thanks for your question! Yes, we have discussed the copyright of repurposing content on forums to create our dataset in Appendix A.3 (3).
>
> To prevent copyright infringement, we follow the Robots Exclusion Protocol of the forum to crawl the content of the posters as our raw data. We also prohibit any secondary distribution of raw data, considering potential licensing and copyright risks. Moreover, as described in Appendix A.2, during our data annotation process, we exclude any content that could violate personal privacy, contain explicit material, depict violence, or involve other sensitive subjects.
>
> Please refer to Appendices A.2 and A.3 (3) for detailed information.

---

### Decision · Program_Chairs · 2024-09-26

**Decision:**

Accept (Spotlight)

**Comment:**

Thank you for the hard work of the reviewers and the detailed responses of the authors. This work has been unanimously recognized by the reviewers, and all reviewers have given a score higher than 7. Therefore, I recommend accepting this paper.